# Identifying biomarkers for diagnosis and disease activity monitoring in PSC-IBD and UC through proteomic profiling: A prospective, biomarker discovery single-center study protocol

Ondrej Fabian [1,2*], Lukas Bajer[3,4], Peter Macinga[3], Jan Brezina[3], Mojmir Hlavaty[3], Pavel Drastich[3], Pavel Wohl[3], Karel Harant[5], Pavel Talacko[5], Eva Sticova[1,6], Andrea Vajsova [1,7], Alena Bohdanecka [8], Filip Tichanek[9], Monika Cahova[8]

1 Clinical and Transplant Pathology Centre, Institute for Clinical and Experimental Medicine, Prague, Czech Republic, 2 Department of Pathology and Molecular Medicine, Faculty of Medicine, Charles University and Thomayer Hospital, Prague, Czech Republic, 3 Department of Gastroenterology and Hepatology, Institute for Clinical and Experimental Medicine, Prague, Czech Republic, 4 Department of Internal Medicine, Motol University Hospital and Second Faculty of Medicine, Charles University, Prague, Czech Republic, 5 Proteomics Core Facility, Faculty of Science, Charles University, Vestec, Czech Republic, 6 Department of Pathology, Royal Vinohrady University Hospital, Srobarova, Prague, Czech Republic, 7 Institute of Pathology of the First Faculty of Medicine and General Teaching Hospital, Studnickova, Prague, Czech Republic, 8 Experimental Medicine Centre, Institute for Clinical and Experimental Medicine, Prague, Czech Republic, 9 Department of Informatics, Institute for Clinical and Experimental Medicine, Prague, Czech Republic

* ondrej.fabian@ikem.cz

**Editor:** Alessandro Granito, University Hospital of Bologna Sant'Orsola-Malpighi Polyclinic Department of Digestive System: Azienda Ospedaliero-Universitaria di Bologna Policlinico Sant'Orsola-Malpighi Dipartimento dell'apparato digerente, ITALY

## Abstract

Inflammatory bowel diseases (IBD) and primary sclerosing cholangitis (PSC) are chronic inflammatory conditions with limited biomarker-driven diagnostic tools. Proteomic profiling offers a promising approach to uncover specific biomarkers that could refine diagnostic accuracy, monitor disease activity, and guide therapeutic strategies. Our primary aim is to identify novel biomarkers for PSC-IBD and conventional ulcerative colitis (UC) via proteomic approach. The secondary aim is to advance the etiopathogenic understanding of the diseases by linking specific proteomic profiles with disease phenotypes. This single-center, prospective, biomarker-discovery study will involve 50 participants with PSC-IBD, 50 with UC, and 50 healthy controls. Biopsy samples from five bowel segments will be analyzed for proteomic signatures by an untargeted approach. The findings will subsequently undergo multi-step external validation in separate cohorts of 30 patients with PSC-IBD and 30 with UC, utilizing targeted proteomics, immunohistochemistry, and ELISA in bowel mucosa and peripheral blood, respectively. This proposed study aims to identify novel biomarkers to improve the diagnostic accuracy of PSC-IBD and UC and refine the disease activity assessment. Its robust design and large sample size provide a strong foundation for successful biomarker identification, with the potential to enhance clinical management of patients.

**Data availability statement:** No datasets were generated or analysed during the current study. All relevant data from this study will be made available upon study completion.

**Funding:** Supported by Ministry of Health, Czech Republic (MH CR) - conceptual development of research organization („Institute for Clinical and Experimental Medicine – IKEM, IN 00023001"), grant no. NW24-05-00168 and no. NU22-06-00269 by MH CR and by the project National Institute for Research of Metabolic and Cardiovascular Diseases (Programme EXCELES, Project No. LX22NPO5104) - Funded by the European Union - Next Generation EU.

**Competing interests:** The authors have declared that no competing interests exist.

## Introduction

Inflammatory bowel diseases (IBD) are chronic systemic inflammatory conditions that primarily affect the gastrointestinal tract (GI) and include Crohn's disease (CD) and ulcerative colitis (UC). Although the exact etiopathogenesis of IBD remains incompletely understood, evidence points to a disrupted interplay between the mucosal immune barrier and the gut microbiome in genetically susceptible individuals. This interaction results in an inappropriate inflammatory response mediated by both innate and adaptive immunity [1]. Increased intestinal epithelial permeability appears to contribute to chronic intestinal inflammation as well. Disruption of intercellular lateral junctions permits a greater proportion of luminal antigens, including bacterial antigens, traverse the epithelium paracellularly, evading degradation into smaller, non-immune peptides, and potentially triggering heightened mucosal immunoreactivity [2,3].

In a subset of patients, IBD is associated with primary sclerosing cholangitis (PSC), a chronic inflammatory disorder of unknown etiology characterized by progressive fibrosing stenosis and obliteration of the extrahepatic and larger intrahepatic bile ducts [4]. The association between PSC and IBD is substantial, with 50% to 80% of PSC patients developing IBD over time [5]. PSC-IBD presents more commonly as pancolitis, as observed in both endoscopic and microscopic evaluations, and carries a fourfold higher risk of colorectal cancer compared to patients with UC alone [6]. The etiopathogenesis of PSC-IBD and its distinctions from UC remain largely unexplored.

Advances in diagnostic techniques and the implementation of modern therapeutic approaches have significantly improved the overall clinical management of patients. However, accurate diagnosis remains a considerable challenge, and many patients continue to face an unfavorable clinical course marked by frequent disease recurrences or severe complications [7]. The variable clinical presentation, inconsistent therapeutic response, and diverse genetic background suggest significant interindividual variability in the pathogenic mechanisms driving disease progression. Consequently, there remains a critical need for novel biomarkers to enhance diagnostic precision, facilitate personalized treatment strategies, accurately assess inflammation severity, and improve the prediction of disease progression and severe complications. Proteomics offers a promising approach for identifying potential new biomarkers through the analysis of a wide array of proteins in tissue samples. However, proteomics studies addressing IBD are limited [8], often constrained by small patient cohorts, and studies specifically focused on PSC-IBD are scarce [9].

The primary goal of this project is to identify new biomarkers to improve diagnostic precision for PSC-IBD and UC, as well as to optimize the assessment of disease activity. The secondary objectives are to enhance our understanding of the etiopathogenesis and immunopathogenic background of PSC-IBD and UC, and to correlate the severity of the histological inflammatory activity and the abundance of specific proteins with the degree of intestinal wall permeability.

## Materials and methods

This is a prospective, observational, single-center biomarker study. The study follows the standard protocols of STROBE guidelines [10] and the SPIRIT reporting guidelines [11].

### Sample size calculation

To estimate the minimum sample size required to detect the difference between groups in terms of proteomic signature, we utilized previously published data on fecal calprotectin, the most conventional non-invasive biomarker for the presence of IBD and disease activity. We identified six studies reporting either median and quartile values or even specific data points. Based on these values, and assuming a log-normal distribution of the fecal calprotectin, we simulated 10 datasets utilizing summary statistics taken from individual studies. All these 10 datasets were used as sources of the two Bayesian models, fitted using the 'brm_multiple' function of the 'brms' R package [12]. One of the models addressed the difference in the Log(Calprotectin) between IBD patients and controls, whereas the second addressed the difference between IBD with lesions vs. IBD without macroscopic lesions. The resulting models were used to simulate new datasets of various sample sizes (employing median posterior values of the group difference and sigma). In each step, the newly generated datasets were subjected to t-test and estimated difference, P-value, non-parametric effect size (Cliff's delta) and its 95% confidence interval were recorded. Power analysis indicated that the sample size of 14–16 is sufficient to achieve 90% for detecting differences in the conventional biomarker between IBD patients and control participants. However, to achieve 90% power for detecting the differences between the IBD subtypes, a sample size of 45–48 patients per group is required. Based on these findings, the study aims to include a minimum of 50 patients per group.

### Study setting and timeline

This project is structured as a sequence of consecutive steps (Fig 1). Initially, candidate biomarkers associated with the diagnosis and/or disease activity will be identified through untargeted proteomic characterization of the bowel mucosa in patients with PSC-IBD and UC. Subsequently, these candidate biomarkers will be validated in external cohorts of PSC-IBD and UC patients, with visualization in the bowel mucosa via targeted proteomics and immunohistochemistry, and in peripheral blood through enzyme-linked immunosorbent assay (ELISA). Targeted proteomic analysis will also be employed to precisely quantify the levels of specific proteins within the tissue.

The project timeline is divided into several consecutive phases. The recruitment phase, including endoscopic sampling of patients and healthy controls started on 01/07/2024 and will be completed by December 2025. In parallel, histopathological evaluation of intestinal samples is being performed during this recruitment phase. Upon completion of sampling, an untargeted proteomic analysis of intestinal mucosal samples will follow, with an anticipated duration of approximately six months, i.e., by March 2026. All data will be obtained and subsequent bioinformatic analysis of proteomic data focused on identifying candidate biomarkers will be performed by June 2026. The publication of the results is expected by December 2026.

In subsequent studies, we plan to verify the predictive significance of the newly identified biomarkers in relation to the long-term clinical outcomes. These future studies will therefore include a long-term follow-up of patients lasting at least two years from the time of biopsy, during which it will be monitored whether the patients develop predefined complications and, if so, after what period of time. The predefined complications will include the need for therapy intensification, hospitalization, surgical intervention, development of Acute Severe Colitis and the occurrence of dysplasia or invasive malignancy.

### Patients' eligibility criteria and recruitment

Based on the number of patients in our center and on the results of power analysis, we anticipate enrolling approximately 50 patients in each discovery cohort. Additionally, 50 individuals with normal endoscopic findings from routine colorectal

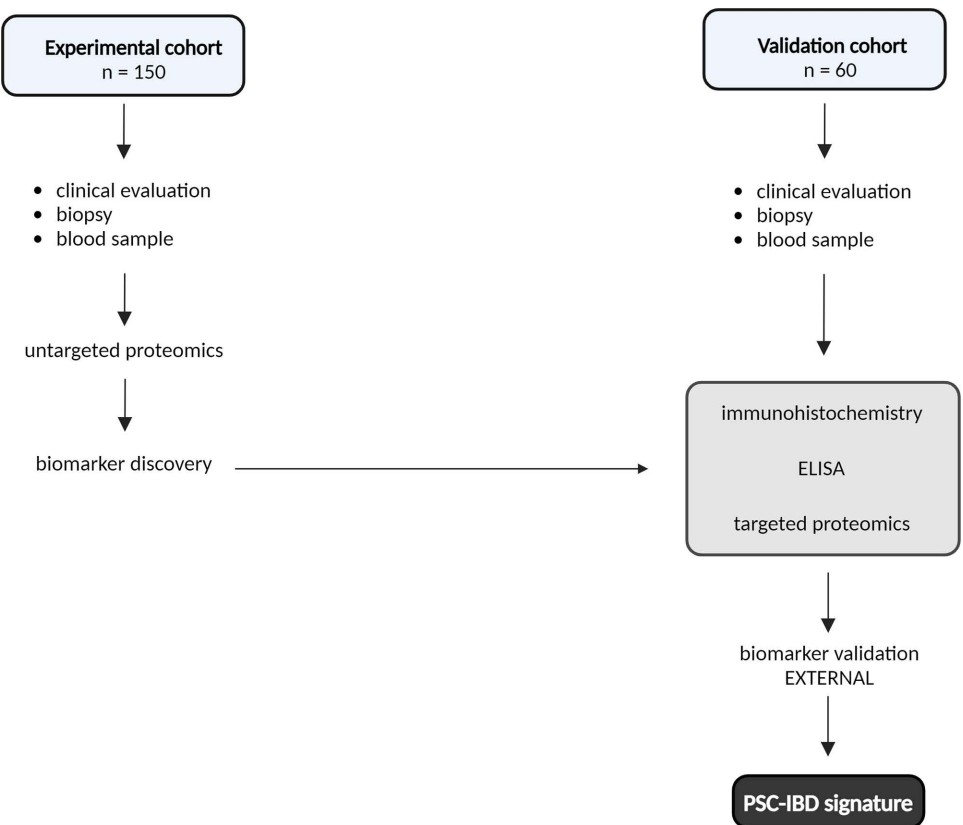

**Fig 1. A flow-chart showing a design of the study.** ELISA = enzyme-linked immunosorbent assay; IBD = inflammatory bowel disease; PSC = primary sclerosing cholangitis.

cancer screening will serve as a cohort of healthy controls. For the validation cohorts, we aim to include 30 patients with PSC-IBD and 30 with UC. The inclusion criteria for PSC-IBD and UC patients in both cohorts are all adults with newly diagnosed or treated PSC-IBD or UC diagnosed in our institution in accordance with official international diagnostic criteria [13–16]. Specifically, the diagnosis of PSC-IBD was established based on characteristic cholangiographic findings of PSC in patients fulfilling diagnostic criteria for IBD. PSC was defined by a persistent cholestatic biochemical profile, typically elevated serum alkaline phosphatase (ALP) and γ-glutamyltransferase levels, and by the presence of multifocal strictures and segmental dilatations of the intra- and/or extrahepatic bile ducts on Magnetic Resonance Cholangiopancreatography (MRCP) or Endoscopic Retrograde Cholangiopancreatography (ERCP), with histological confirmation when available. Secondary causes of sclerosing cholangitis, such as biliary obstruction, IgG4-related disease or infectious cholangitis, were excluded. UC was diagnosed according to standard clinical, endoscopic and histopathological criteria, characterized by continuous mucosal inflammation starting from the rectum and extending proximally, with typical microscopic features of chronic mucosal injury and absence of granulomatous or transmural inflammation. Other causes of colitis, including infectious, ischemic or drug-induced forms, were ruled out by appropriate microbiological, serological and histological investigations. All patients will be enrolled irrespective of their current therapy regimen. The exclusion criteria are the presence of dysplasia or invasive cancer in histology and the presence of acute intestinal infection. For healthy controls, the inclusion criteria are normal endoscopic and histological findings during the colorectal cancer endoscopic screening, while the presence of endoscopic or microscopic inflammation, and the presence of dysplasia or cancer excludes the individuals from the study.

## Data collection

**Collection of bioptic samples and peripheral blood.** For each patient, a standard bioptic sampling from five bowel segments (terminal ileum, cecum, transverse colon, descending colon and rectum) is performed during the endoscopic examination. The tissue is fixed in 10% formol and transported to the pathology department. The samples are routinely processed and embedded in paraffin blocks. After that, 3 µm thick sections are cut, stained with hematoxylin and eosin, and evaluated by a gastrointestinal pathologist blinded to the clinical data. Additional native, non-fixed biopsy samples from three bowel segments (terminal ileum, cecum and rectum) are collected, frozen in liquid nitrogen at −80°C, and prepared for subsequent proteomic analysis. Apart from that, 3 ml of peripheral blood is collected from each patient, with serum separated and aliquoted into 250 µl portions. All samples are stored at −80°C for further analysis.

**Intestinal wall permeability assessment.** Concentration of circulating markers well established for the detection of gut-barrier dysfunction (e.g., zonulin, i-FABP, L-FABP, LPS, citruline, DAO) [17] will be assessed by ELISA.

**Estimation of systemic inflammation.** Basic inflammatory biomarkers, including C-reactive protein (CRP), platelet count, erythrocyte sedimentation rate (ESR), and albumin levels, are measured using standard laboratory methods. Additionally, IBD-associated antibodies, such as anti-Saccharomyces cerevisiae antibodies (ASCA) and perinuclear anti-neutrophil cytoplasmic antibodies (pANCA), are analyzed using commercially available kits. Fecal calprotectin levels are also assessed as a marker of intestinal inflammation.

**Disease activity assessment.** Disease activity is determined at the microscopic, endoscopic, and clinical levels using appropriate scoring indices. Clinical activity is assessed using the Mayo Score, endoscopic activity with the Mayo Endoscopic Subscore, and histological activity with the Nancy Histological Index [18,19]. Monitoring of PSC activity will be performed based on findings from MRCP or ERCP and liver function tests, specifically ALP levels.

**Proteomic analysis of the native samples.** The frozen samples will be mechanically homogenized on cryo-mill, dissolute in buffer containing detergent, and subjected to trypsin digestion. An untargeted Data Independent Acquisition approach will be used to measure the resulting peptides. Mass spectrometry data will be acquired on the high-end Mass spectrometer Orbitrap Ascend coupled with nanoLC Ultimate 3000. Peptides from raw mass spectrometric data will be identified and quantified by the proteomic software MaxQuant [20]. Results will be presented as intensity for each identified protein in each tissue sample in the dataset.

Candidate proteins for biomarkers of PSC-IBD and UC will be selected from the resulting data. Peptides suitable for absolute by targeted parallel reaction monitoring approach quantification will be selected for each candidate protein. The targeted approach offers superior quantitative and qualitative accuracy in comparison to the untargeted approach and is the most suitable choice for proteomics validation of possible biomarkers from the previous step on the samples from validation cohort of the patients.

**The immunochemical verification of the candidate biomarkers.** For the immunohistochemical evaluation of bowel tissue samples, 4 µm-thick sections will be prepared from the paraffin blocks and processed using the Ventana Benchmark ULTRA automated staining system (Roche Diagnostics). Monoclonal and polyclonal antibodies targeting the candidate proteins will be applied for staining. The reactions will be visualized by Ultraview Detection System (Ventana Medical Systems). The slides will be counterstained with hematoxylin. If the target protein is positively identified in the tissue through immunohistochemistry, the degree of positivity will be graded based on intensity (weak, moderate, or strong) and extent (focal or diffuse). Candidate biomarker levels in peripheral blood will be measured using ELISA, with each sample tested in triplicate to minimize potential intra-assay variability.

## Statistical analysis

All statistical analyses will be conducted using R software [21] and the scripts will be published on GitHub. Spearman's correlation will evaluate potential relationships between variables.

Proteomic analyses, which generate large volumes of complex data, will require multivariate statistical evaluation. Prior further analyses, the lrSVD method, which should efficiently estimate zeros (replace them for non-zero numbers) even in very sparse compositional datasets using global data patterns, will be applied to all proteomic datasets. Principal Component Analysis (PCA) will be applied to visualize trends in spectral clustering and identify outliers. Group differences in protein levels will be analyzed using PERMANOVA via the 'vegan' R package [22]. Both PCA and PERMANOVA will employ Euclidean distances calculated after log2 transformation and Z-standardization of protein levels.

To identify proteins showing strongest association with specific diagnoses or disease activity, a series of linear models will be fitted to log2-transformed protein levels, each model focusing on an individual protein. The false discovery rate (FDR) will be controlled using the Benjamini-Hochberg correction. Proteins significantly associated with diagnosis or disease activity after FDR correction will be further analyzed in a validation cohort to assess generalizability.

To assess whether omics profiles contains a reproducible signal distinguishing diagnostic or disease activity groups, we will use logistic or ordinal elastic net regression models ($\alpha = 0.5$) via the 'glmnet' [23] and 'ordinalNet' [24] R packages, with all features log2-transformed and Z-standardized. Ten-fold cross-validation will be used on the training data to identify the optimal penalty parameter $\lambda$, selecting the most regularized model whose cross-validated deviance is within one standard error of the minimum. To ensure robust internal validation and prevent model overfitting, predictive performance will be evaluated using out-of-bag bootstrap resampling with nested cross-validation (500 iterations) to estimate out-of-sample performance (area under the ROC curve, AUC) and prediction stability. During this internal validation procedure, each iteration will involve: (i) drawing a bootstrap sample of subjects (with replacement); (ii) re-running model tuning (selection of $\lambda$ using cross-validation) within the bootstrap sample; (iii) fitting the model on the bootstrap sample and evaluating its performance (ROC-AUC) on out-of-bag subjects (those not included in the given bootstrap sample). The median, 2.5th, and 97.5th percentiles of the bootstrap AUC distribution will summarize model discrimination and its uncertainty across 500 bootstrap iterations.

Elastic net regression will then be repeated with the top five proteins showing the largest estimated effects (indicating the highest predictive potential) to evaluate the loss of performance when focusing on a narrower subset of proteins. Prediction models will also be validated in an independent cohort to assess their generalizability beyond the development dataset.

### Ethics approval and consent to participate

The study is conducted according to the guidelines of the Declaration of Helsinki and approved by the The Ethics Committee of the Institute for Clinical and Experimental Medicine and Thomayer University Hospital (protocol code 16571/23; G-23–33). Written informed consent is obtained before enrolment for the study in each subject. The investigator will make sure that subjects comprehend the nature of the study, the study procedures, and the risks and benefits of participation. There will be no penalty if a participant decides to withdraw from the study before it ends. Study progress, data integrity, and ethical and safety concerns will be reviewed by the investigative team monthly (and more frequently if needed). Existing security provisions in accordance with GDPR and Charles University institutional security rules for the protection of personal identifying information are maintained. These measures include training of personnel, control of access to office space, and electronic security provisions.

### Patient and public involvement

There has been no patient and public involvement.

### Discussion

IBD were traditionally regarded as conditions limited to the GI tract, with CD and UC recognized as the two primary phenotypes. However, they are now perceived as systemic disorders with a strong predilection for the GI tract. Their phenotypes are considered part of a continuous spectrum of conditions, ranging from CD involving the small bowel and upper GI

tract, through colonic CD and IBD-unclassified, to distal UC [13,25]. As previously mentioned, the precise etiopathogenesis remains incompletely understood. Nonetheless, an imbalance in homeostasis between the intestinal epithelial barrier, immune barrier, and commensal intestinal microbiota in a genetically susceptible host is considered a key mechanism in the disease's onset and progression [1]. Inflammation is primarily driven by T cells, with T helper 1 and 2 subpopulations playing dominant roles. Recent studies have also demonstrated significant involvement of Th17 cells, natural killer cells, and antigen-presenting dendritic cells [26]. Additionally, the role of impaired mucosal integrity is increasingly recognized as a contributing factor. The intestinal lumen is lined by a single layer of columnar epithelial cells bound by tight junctions, adherent junctions, and desmosomes [27]. Under normal conditions, up to 90% of all antigens are transported transcellularly from the intestinal lumen into the bloodstream, with associated degradation into small, non-immunogenic peptides. Only 10% of antigens are transported paracellularly, preserving their antigenic properties and playing an essential role in inducing immune tolerance [2]. Increased intestinal permeability allows a larger proportion of immunologically active antigens to translocate into the bloodstream, potentially contributing to the risk of various immunopathological diseases, including IBD [3].

Over the past decades, advancements in the understanding of pathogenesis, morphology, and clinical presentation have resulted in increasingly refined subclassifications of IBD. Diagnosis now requires a multidisciplinary approach, integrating the clinical course of the disease with laboratory, radiological, endoscopic, and histopathological findings [14,18]. Despite these advancements, up to 6% of adult and 13% of pediatric IBD cases remain unclassified [7]. Accurate diagnosis of CD or UC, along with precise phenotypic subclassification, is essential for guiding optimal medical and surgical management. This underscores the ongoing effort to identify novel biomarkers that not only enhance diagnostic precision and enable a more accurate assessment of disease activity, but also help to stratify patients according to disease behavior, predict therapeutic response, and guide personalized treatment decisions. Biomarkers reflecting immune activation, epithelial barrier dysfunction, or fibrosis could assist in identifying patients at higher risk of aggressive or treatment-refractory disease. These advancements are expected to facilitate the earlier initiation of appropriate therapy, improve therapeutic monitoring, and ultimately reduce the long-term incidence of severe complications. Numerous promising serum, fecal, and tissue biomarkers have been identified over time, though only a small fraction have been integrated into routine clinical practice. An ideal biomarker would be easily detectable by non-invasive methods, highly sensitive, specific, and cost-effective. Currently, no biomarker fulfills all these criteria. Among the most widely used are several conventional serological markers such as ASCA and pANCA, yet their clinical utility is restricted by suboptimal sensitivity and specificity. Notably, ASCA positivity has also been observed in non-IBD conditions, including untreated celiac disease [28], where it likely reflects a nonspecific immune response to microbial antigens associated with increased intestinal permeability. This overlap emphasizes that ASCA or pANCA positivity alone should not be interpreted as definitive evidence of IBD, particularly in patients with atypical or overlapping clinical features. In this study we seek to overcome these limitations by identifying biomarkers with greater disease specificity and a closer mechanistic link to intestinal inflammation and mucosal barrier dysfunction.

As noted above, IBD are often tied to PSC, which is a debilitating disease with many patients progressing to liver cirrhosis and requiring liver transplantation. The connection between PSC and IBD is well established, with most PSC patients developing IBD during their lifetime [5]. PSC-associated IBD has traditionally been classified as UC and, less commonly, as CD. However, it is now increasingly recognized as a distinct form of IBD, and many specialists avoid further subclassification when the disease occurs in the context of PSC. Unlike conventional UC, PSC-IBD more commonly presents as pancolitis on both endoscopic and microscopic examination. Morphological changes typically involve the right colon with a characteristic reverse gradient of inflammation, common occurrence of backwash ileitis, and frequent sparing of the rectum [29]. Importantly, the condition carries a substantially higher risk of colorectal cancer compared to conventional IBD, despite a milder clinical course [6]. It is therefore clinically imperative to distinguish PSC-associated IBD from conventional IBD, a distinction further emphasized by the potential for biliary disease to manifest long after the onset of

intestinal inflammation [30]. Unfortunately, no reliable biomarker has been identified to distinguish PSC-associated IBD from conventional IBD, apart from the presence of advanced biliary disease.

When searching for new reliable biomarkers of a disease or specific clinical condition, it is essential to determine whether the biomarker should be identified at the level of DNA, transcriptomics, or protein expression. Genetic predisposition clearly plays an essential role in the etiology of IBD, and advances in molecular genetics, particularly genome-wide association studies, have identified multiple candidate genes whose pathogenic mutations or single nucleotide polymorphisms correlate with an increased risk of CD or UC or with a higher likelihood of aggressive disease phenotypes. However, genetic alterations do not always correlate with downstream protein expression. Additionally, environmental factors play a significant role in IBD etiology, impacting transcriptomic and protein expression without affecting the DNA level. Biomarker discovery at the protein level is therefore more clinically relevant, as proteins directly represent functional and dynamic pathological processes, such as inflammation and fibrosis, offering valuable insights into disease activity and progression. In contrast, DNA-based biomarkers, while valuable for understanding genetic predispositions, are static and may not accurately reflect the current disease state. Proteomics offers a dynamic approach, linking genomic and transcriptomic changes to the phenotypic manifestations of IBD, providing insights into the mechanisms driving disease initiation and progression. Consequently, proteomics represents an ideal approach for identifying extensive sets of candidate biomarkers by enabling broad-spectrum protein analysis from tissue samples or body fluids [31]. Proteomic studies in IBD remain limited, often involving small patient cohorts, with only a few identified biomarkers advancing to clinical application. A 2019 review by Assadsangabi et al. [8] highlighted only 19 studies, of which 13 analyzed tissue samples (endoscopic biopsies or resection specimens), while the remainder focused on body fluids. Even more limited information is available on proteomics in PSC-IBD. To our knowledge, the only relevant study by Vessby et al. [9] analyzed nine PSC-IBD patients and seven UC patients in clinical and histologic remission, and seven healthy controls, identifying AGPAT1 protein as a potential PSC-IBD marker.

From a practical point of view, the potential clinical implementation of the identified biomarkers will also depend on their cost-effectiveness and accessibility. Although proteomic discovery phases rely on advanced and resource-intensive technologies, subsequent validation using immunohistochemistry or ELISA offers a cost-efficient and scalable means for clinical adoption. By focusing on biomarkers measurable through routine laboratory techniques, this study aims to facilitate the development of affordable diagnostic tools that can be integrated into everyday clinical practice without substantial increases in healthcare costs.

In conclusion, our proposed study aims to identify novel biomarkers to enhance the diagnostic accuracy for PSC-IBD and UC and improve assessments of the disease activity. This could enable better risk stratification and personalized treatment plans. Our study realistically anticipates including 150 subjects, making it one of the largest IBD proteomic studies to date. We believe the study is well-founded, timely, and serves a meaningful purpose.

## Supporting information

**S1 File. SPIRIT checklist Fabian.**
(DOCX)

## Author contributions

**Conceptualization:** Ondrej Fabian, Filip Tichanek, Monika Cahova.

**Data curation:** Lukas Bajer, Peter Macinga, Jan Brezina, Mojmir Hlavaty, Pavel Drastich, Pavel Wohl, Karel Harant, Pavel Talacko, Eva Sticova, Andrea Vajsova, Monika Cahova.

**Formal analysis:** Filip Tichanek, Monika Cahova.

**Funding acquisition:** Ondrej Fabian, Monika Cahova.

**Investigation:** Lukas Bajer, Peter Macinga, Jan Brezina, Mojmir Hlavaty, Pavel Drastich, Pavel Wohl, Pavel Talacko, Eva Sticova, Andrea Vajsova, Alena Bohdanecka.

**Methodology:** Ondrej Fabian, Lukas Bajer, Karel Harant, Pavel Talacko, Alena Bohdanecka, Filip Tichanek, Monika Cahova.

**Project administration:** Monika Cahova.

**Software:** Karel Harant, Pavel Talacko, Alena Bohdanecka, Filip Tichanek, Monika Cahova.

**Supervision:** Ondrej Fabian, Monika Cahova.

**Visualization:** Monika Cahova.

**Writing – original draft:** Ondrej Fabian, Karel Harant, Monika Cahova.

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
