## [Decision Letter · Decision Letter 0]

4 Nov 2025

Dear Dr. Fabian,

Thank you for submitting your manuscript to PLOS ONE. After careful consideration, we feel that it has merit but does not fully meet PLOS ONE’s publication criteria as it currently stands. Therefore, we invite you to submit a revised version of the manuscript that addresses the points raised during the review process.

We look forward to receiving your revised manuscript.

Kind regards,

Alessandro Granito

Academic Editor

PLOS ONE

Journal Requirements:

“Supported by Ministry of Health, Czech Republic (MH CR) - conceptual development of research organization (“Institute for Clinical and Experimental Medicine – IKEM, IN 00023001“), grant no. NW24-05-00168 and no. NU22-06-00269 by MH CR and by the project National Institute for Research of Metabolic and Cardiovascular Diseases (Programme EXCELES, Project No. LX22NPO5104) - Funded by the European Union - Next Generation EU.”

Reviewers' comments:

Reviewer's Responses to Questions

**Comments to the Author**

1. Does the manuscript provide a valid rationale for the proposed study, with clearly identified and justified research questions?

Reviewer #1: Yes

Reviewer #2: Yes

2. Is the protocol technically sound and planned in a manner that will lead to a meaningful outcome and allow testing the stated hypotheses?

Reviewer #1: Yes

Reviewer #2: Yes

3. Is the methodology feasible and described in sufficient detail to allow the work to be replicable?

Reviewer #1: Yes

Reviewer #2: Yes

4. Have the authors described where all data underlying the findings will be made available when the study is complete?

Reviewer #1: Yes

Reviewer #2: Yes

5. Is the manuscript presented in an intelligible fashion and written in standard English?

Reviewer #1: Yes

Reviewer #2: Yes

You may also provide optional suggestions and comments to authors that they might find helpful in planning their study.

Reviewer #1: The study protocol, "Identifying biomarkers for diagnosis and disease activity monitoring in PSC-IBD and UC through proteomic profiling," is highly relevant. Still, its single-centre, discovery nature presents translational risks that must be mitigated.

Recommendations for the Protocol

1) Validation and Generalisability:'Given that the study is single-centre, the protocol must explicitly acknowledge the need for future external validation' in an independent cohort. To strengthen the discovery phase, the statistical analysis plan should include a robust 'internal validation strategy', such as nested cross-validation, to prevent overfitting of the protein signature.

2) Standardised Phenotyping:'Biomarker results rely entirely on accurate clinical context. The protocol must mandate the use of the 'gold-standard measures' for defining disease states:

For UC activity and remission, use the 'Mayo Endoscopic Subscore' and a 'Histological Scoring System'(e.g., Nancy Index).

For PSC activity, use 'MRCP'(Magnetic Resonance Cholangiopancreatography) findings and serial 'serum Alkaline Phosphatase (ALP)' levels.

3) Clinical Endpoints: The predictive utility of any biomarker panel must be defined. The protocol should clearly state the 'long-term clinical outcomes' (e.g., treatment escalation, surgery, malignancy development) that the proteomic signatures will be tested to predict over the follow-up period.

Reviewer #2: In this study, Ondrej Fabian and colleagues, aimed to identify novel proteomic biomarkers for PSC-IBD and UC by means an untargeted proteomic analysis followed by multi-step validation in independent cohorts. The relatively large sample size (150 subjects in the discovery cohort) is commendable and addresses a common limitation in previous IBD proteomic studies. The study's focus on correlating proteomic profiles with disease phenotypes and histological activity is also promising, potentially leading to a more nuanced understanding of disease pathogenesis and improved diagnostic accuracy. The rationale for identifying more reliable biomarkers is clear: current diagnostic tools for IBD have limitations in accuracy and disease activity assessment, leading to delayed diagnoses and suboptimal treatment strategies.

However, the study's clinical relevance could be further emphasized by explicitly addressing the shortcomings of currently available serological markers. While the study mentions assessing ASCA and pANCA, it needs to delve deeper into the limitations of these markers. For example, the lack of sensitivity and specificity that can overlap IBD or non-IBD conditions are key. More importantly, one critical point that should be recalled and discussed, is the frequent detection of ASCA in untreated celiac disease, as previously demonstrated (doi: 10.1111/j.1365-2036.2005.02417.x), suggesting that serum ASCA production might be a non-specific response to microbial antigens due to increased intestinal permeability. This phenomenon is particularly relevant as both celiac disease and IBD share similar clinical presentations and can sometimes be misdiagnosed, especially in early stages where both diseasorders may present ASCA positivity without clinical signs (Gut. 2006;55(2):296. ).

Therefore, the study should include a thorough discussion of these findings, emphasizing that ASCA positivity alone should not be considered a definitive marker for IBD, particularly in patients with atypical symptoms.

-The authors should clearly define the criteria used to differentiate PSC-IBD from UC and other forms of colitis. Given the phenotypic overlap, stringent diagnostic criteria are essential to ensure accurate biomarker identification.

Longitudinal Assessment: While the study is prospective, it would be valuable to incorporate a longitudinal component to assess the biomarkers' ability to predict disease progression, response to therapy, and risk of complications.

-Cost-Effectiveness: The study should address the potential cost-effectiveness of the identified biomarkers in clinical practice. While novel biomarkers may improve diagnostic accuracy, their implementation depends on affordability and accessibility.

-Patient stratification. It might be good to mention the biomarkers that might help discriminate the disease behavior, and select different treatment for the patients.

**Do you want your identity to be public for this peer review?** For information about this choice, including consent withdrawal, please see our Privacy Policy

Reviewer #1: **Yes:**  Rajarajan Ramamoorthy

Reviewer #2: No

---

## [Author Response · Author response to Decision Letter 1]

24 Nov 2025

Dear Reviewers,

The authors have addressed all of your comments and incorporated them into the manuscript. The revised manuscript has been uploaded in both the marked and unmarked versions, as requested by the editorial team. Our detailed responses to your requirements and comments on the text can be found in the uploaded document reviewers_answers.doc.

Kind regards,

Ondrej Fabian, on behalf of all co-authors

---

## [Decision Letter · Decision Letter 1]

15 Jan 2026

Identifying biomarkers for diagnosis and disease activity monitoring in PSC-IBD and UC through proteomic profiling: a prospective, biomarker discovery single-center study protocol

PONE-D-25-49137R1

Dear Dr. Fabian,

We’re pleased to inform you that your manuscript has been judged scientifically suitable for publication and will be formally accepted for publication once it meets all outstanding technical requirements.

Kind regards,

Alessandro Granito

Academic Editor

PLOS One

Additional Editor Comments (optional):

Reviewers' comments:

Reviewer's Responses to Questions

**Comments to the Author**

1. Does the manuscript provide a valid rationale for the proposed study, with clearly identified and justified research questions?

Reviewer #1: Yes

Reviewer #2: Yes

2. Is the protocol technically sound and planned in a manner that will lead to a meaningful outcome and allow testing the stated hypotheses?

Reviewer #1: Yes

Reviewer #2: Yes

3. Is the methodology feasible and described in sufficient detail to allow the work to be replicable?

Reviewer #1: Yes

Reviewer #2: Yes

4. Have the authors described where all data underlying the findings will be made available when the study is complete?

Reviewer #1: Yes

Reviewer #2: Yes

5. Is the manuscript presented in an intelligible fashion and written in standard English?

Reviewer #1: Yes

Reviewer #2: Yes

You may also provide optional suggestions and comments to authors that they might find helpful in planning their study.

Reviewer #1: I am grateful to the authors for their thorough response to my previous comments. The revisions have successfully addressed the concerns regarding the clinical rigour and translational potential of the protocol. By explicitly acknowledging the single-centre nature of the study and introducing an external validation plan with independent cohorts, the authors have significantly improved the reliability of their findings. I also welcome the inclusion of gold-standard measures for disease phenotyping, such as the Mayo Endoscopic Subscore and Nancy Index for Ulcerative Colitis, as well as MRCP and serial Alkaline Phosphatase levels for Primary Sclerosing Cholangitis.

Furthermore, the expanded discussion on long-term clinical endpoints and cost-effectiveness provides a much clearer path for the future clinical use of these biomarkers.

I have some optional suggestions that the authors may find useful as they begin their data collection. To ensure the highest quality of proteomic data, it may be beneficial to standardise the timing of sample collection and document the 'cold ischaemia time' between tissue resection and freezing.

Additionally, establishing a multicentre network or biobank partnership early in the process would facilitate a smoother transition to the external validation phase. These proactive steps would further enhance the long-term impact of this important research.

Reviewer #2: The revised manuscript contains the necessary additions and it has now been satisfactorily improved as requested. I have no further comment and the manuscript can be accepted in its current form.

**Do you want your identity to be public for this peer review?** For information about this choice, including consent withdrawal, please see our Privacy Policy

Reviewer #1: **Yes:**  Rajarajan Ramamoorthy

Reviewer #2: No

---

## [Editor Report · Acceptance letter]

PONE-D-25-49137R1

PLOS One

Dear Dr. Fabian,

I'm pleased to inform you that your manuscript has been deemed suitable for publication in PLOS One. Congratulations! Your manuscript is now being handed over to our production team.

Kind regards,

on behalf of

Dr. Alessandro Granito

Academic Editor

PLOS One